# HA-MOP knockin mice express the canonical μ-opioid receptor but lack detectable splice variants

Sebastian Fritzwanker[1], Lionel Moulédous [2], Catherine Mollereau [2,3], Carine Froment[3], Odile Burlet-Schiltz[3], Felix Effah [4], Alexis Bailey [4], Mariana Spetea[5], Rainer K. Reinscheid[1], Stefan Schulz [1✉] & Andrea Kliewer [1✉]

G protein-coupled receptors (GPCRs) are notoriously difficult to detect in native tissues. In an effort to resolve this problem, we have developed a novel mouse model by fusing the hemagglutinin (HA)-epitope tag sequence to the amino-terminus of the μ-opioid receptor (MOP). Although HA-MOP knock-in mice exhibit reduced receptor expression, we found that this approach allowed for highly efficient immunodetection of low abundant GPCR targets. We also show that the HA-tag facilitates both high-resolution imaging and immunoisolation of MOP. Mass spectrometry (MS) confirmed post-translational modifications, most notably agonist-selective phosphorylation of carboxyl-terminal serine and threonine residues. MS also unequivocally identified the carboxyl-terminal [387]LENLEAETAPLP[398] motif, which is part of the canonical MOP sequence. Unexpectedly, MS analysis of brain lysates failed to detect any of the 15 MOP isoforms that have been proposed to arise from alternative splicing of the MOP carboxyl-terminus. For quantitative analysis, we performed multiple successive rounds of immunodepletion using the well-characterized rabbit monoclonal antibody UMB-3 that selectively detects the [387]LENLEAETAPLP[398] motif. We found that >98% of HA-tagged MOP contain the UMB-3 epitope indicating that virtually all MOP expressed in the mouse brain exhibit the canonical amino acid sequence.

[1] Department of Pharmacology and Toxicology, Jena University Hospital—Friedrich Schiller University Jena, Jena, Germany. [2] Research Center on Animal Cognition, Center for Integrative Biology, CNRS, UPS, Toulouse University, Toulouse, France. [3] Institut de Pharmacologie et de Biologie Structurale (IPBS), CNRS, UPS, Université de Toulouse, Toulouse, France. [4] Pharmacology Section, St George's University of London, London, UK. [5] Department of Pharmaceutical Chemistry, Institute of Pharmacy and Center for Molecular Biosciences Innsbruck (CMBI), University of Innsbruck, Innsbruck, Austria. ✉email: stefan.schulz@med.uni-jena.de; andrea.kliewer@med.uni-jena.de

Since the initial cloning of the single-copy μ-opioid receptor gene (OPRM1), numerous isoforms resulting from alternative mRNA splicing have been postulated in humans, rats, and mouse[1–3], however, their in vivo significance is largely unknown. While at least 15 alternative MOP transcripts involve carboxyl-terminal (C-terminal) variants, others are proposed to form non-canonical receptors that no longer conform to the 7-transmembrane topology of classical GPCRs[4,5]. Most transcripts have been solely identified at the RNA level and studies into their functional significance usually employed in vitro models of recombinantly-expressed receptor constructs[6,7]. Only a few investigations ever attempted to verify the presence of MOP isoforms at the protein level. For example, using polyclonal antisera expression of two C-terminal MOP splice variants (MOP1B4 and MOP1C) could be visualized in various areas of the mouse brain[3,8,9]. Using a similar approach, the MOP1B splice variant of the rat was detected in discrete brain areas[10]. However, in none of these studies, polyclonal antisera were rigorously tested in knockout tissues.

At the functional level, several attempts were made to match specific isoforms with particular opioid compounds or link them to selected physiological opioid effects. These studies mostly used isoform-specific RNA knock-down approaches, suffering from inherent limitations of unknown efficacy while working against a background of abundantly expressed canonical receptors. One study described co-expression of the murine MOP1D isoform together with gastrin-releasing peptide receptors in lamina 1 of the spinal cord and their co-involvement in morphine-induced itch[11]. To date, at least 30 C-terminal MOP splice variants have been postulated (mouse, rat, human) based on mRNA sequencing data, out of which 15 are of murine origin. Noteworthy, 11 of these 15 murine transcripts have been published by a single lab[12–14].

In general, the lack of validated high-affinity antibodies and the low abundance of MOP protein expression in native tissues have made it difficult to conclusively verify the existence of postulated variant MOP isoforms. Earlier attempts to address this problem involved a transgenic model expressing a fusion of red fluorescent protein to the C terminus of MOP, harboring the risk of interference with intracellular signaling[15]. Hemagglutinin (HA)-MOP transgenic mice overcome these limitations and allow for the first time isolation of native MOP proteins from tissues. By design of the transgenic construct, HA-MOP allows for the identification of any C-terminal variants. In this study, we used mass spectrometry (MS) to directly identify MOP C-terminal variants and differential agonist-dependent phosphorylation states of native HA-MOP receptors. Our results suggest that C-terminal variants of MOP either do not exist or occur at such low abundance that their physiological significance is questionable.

## Results

### Characterization of HA-MOP knockin mice

We generated a novel transgenic mouse model by fusing the HA-epitope tag sequence YPYDVPDYA to the amino-terminus of MOP (HA-MOP) (Fig. 1a and Supplementary Fig. 1). HA-MOP knockin mice were viable and healthy and showed no differences in body weight and basal pain response (Supplementary Fig. 2d, e). Binding affinity (Kd) of [3H]DAMGO determined in saturation binding studies using brain membranes was not changed between wild-type (WT) and HA-MOP mice (Supplementary Fig. 2a). However, the total number of receptors ($B_{max}$) was significantly decreased by about 50% in the brain from HA-MOP mice (Supplementary Fig. 2b). Similar results were obtained by quantitative autoradiography of [3H]DAMGO binding in the nucleus accumbens, thalamus, hypothalamus, amygdala, and somatosensory cortex (Supplementary Fig. 2c). Furthermore, DAMGO-stimulated [35S]GTPγS binding is significantly decreased in HA-MOP brain membranes compared to wild-type, both in potency ($EC_{50}$) and efficacy (% stimulation). Accordingly, HA-MOP knockin mice exhibit reduced antinociceptive responses to morphine, which are similar to heterozygous MOP knockout ($MOP^{+/-}$) mice (Supplementary Fig. 2f). In order to confirm MOP expression in native mouse tissues, we first immunoprecipitated HA-MOP from whole-brain lysates using anti-HA magnetic beads (Fig. 1b). HA-MOP was detected in western blots using both an anti-HA antibody directed to the N-terminal tag (Fig. 1b, left panel) as well the well-characterized UMB-3 antibody directed to the carboxyl-terminal tail of the canonical MOP receptor (Fig. 1b, right panel), while no band was detected in WT mice. Next, we evaluated HA-MOP expression levels in different organs (Fig. 1c). The highest levels of HA-MOP expression were found in the brain and spinal cord, as well as in the gastrointestinal tract. Furthermore, we also detected moderate HA-MOP expression in the heart, thymus, spleen, and uterus. HA-MOP expression was not detectable in lysates prepared from lung, liver, pancreas, or kidneys. We then prepared brain slices from adult mice and examined HA-MOP expression in different brain regions using immunohistochemistry (Figs. 1 and 2). As shown in Fig. 1d, immunostaining for HA-MOP was detected in the striatum (CPu) of knockin mice but not in WT mice. At high resolution, HA-MOP was detectable on habenula neuronal cell bodies, as well as cell bodies and dendrites of amygdala neurons. Next, we analyzed HA-MOP expression at different developmental stages. The HA-MOP was first detectable at E16 and expression increases until birth. Slices from postnatal mice at P7 and P10 already show the typical MOP expression pattern of adult striatum.

### MOP phosphorylation pattern in vivo

Agonist-dependent MOP phosphorylation plays a major role in receptor desensitization and downstream signaling. To analyze the different phosphorylation patterns (involving S375, T370, T376, and T379) in native tissues, we administered various clinically relevant MOP agonists including the MOP antagonist naltrexone and saline as controls (Fig. 3). All MOP agonists induced strong MOP phosphorylation at S375 detected using phosphosite-specific antibodies. However, the MOP partial agonists morphine and oxycodone were not able to promote higher-order phosphorylation at additional phospho-acceptor sites. All high-efficacy agonists such as methadone, fentanyl, sufentanil, or etonitazene promoted robust receptor phosphorylation at T370, T376, and T379 in addition to S375. No signals were detected after naltrexone or saline treatment. The MS analysis confirmed that peptides from the HA-MOP carboxyl-terminus were phosphorylated in the agonist-stimulated conditions, but not in the control conditions (Table 1). The peptide 349EFCIPTSSTIEQQNSAR365 phosphorylated on S363 was identified in samples from mice treated with fentanyl and etonitazene. Single phosphorylation on T370 and double phosphorylation on T370/S375 or T370/T376, were unequivocally identified on the peptide 366IRQNTREHPSTANTVDR382 and the shorter form 368QNTREHPSTANTVDR382 in the three agonist-treated conditions. However, the triple phosphorylation at T370/S375/T376 was only observed after treatment with the high-efficacy agonist etonitazene. These results confirm and extend previous in vitro and in vivo studies demonstrating hierarchical and agonist-selective phosphorylation of the MOP carboxyl-terminal tail, indicating that the HA-MOP mouse model is a valid tool for biochemical characterization of endogenously expressed μ-opioid receptors.

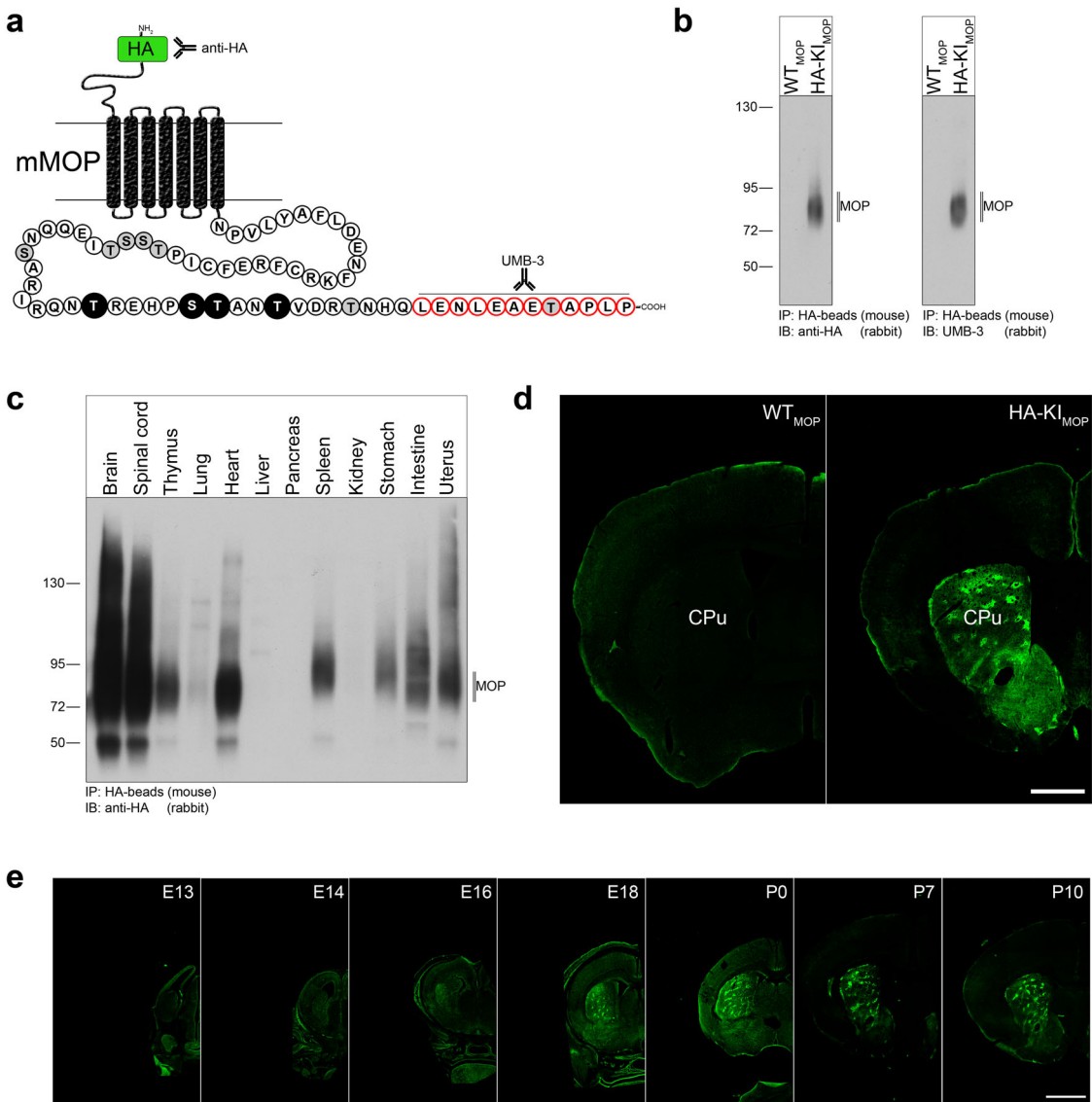

**Fig. 1 Localization and phosphorylation of HA-MOP in mouse brain. a** Schematic representation of the N-terminally tagged HA-MOP and the C-terminal tail with potential phosphorylation sites (black: antibody detectable phosphorylation sites; light gray: possible phosphorylation sites). **b** Western blot analysis of brain lysates from HA-MOP mice compared to wild-type animals after immunoprecipitation with anti-HA magnetic beads. Proteins were either immunoblotted with anti-HA antibody (left panel) or UMB-3 (right panel). **c** Western blot analysis of HA-MOP expression in different organs. Blots are representative of three independent experiments. **d** 40 μm brain section of an adult HA-MOP mouse (right panel) compared to wild-type (left panel). Slices were stained with Cy3-labeled (550 nm) secondary antibody and pseudo-colored afterward. **e** Developmental expression of HA-MOP in mouse striatum across different age stages. 40 μm brain slices were stained with anti-HA antibody followed by Cy3-labeled secondary antibody and pseudo-colored afterward. **d**, **e** Scale bar = 1000 μm. CPu caudate putamen.

**Investigation of alternative MOP splice variants in vivo**. In order to identify peptides belonging to the different putative MOP isoforms, anti-HA-immunoprecipitated samples from control and three agonist-treated conditions were subjected to SDS-PAGE separation. HA-MOP corresponding bands were in-gel digested for further analysis by nano–liquid chromatography-tandem MS (nanoLC–MS/MS). NanoLC–MS/MS identified 4 peptides located in the carboxyl-terminal domain of the MOP receptor, covering positions 349 to 398 of the canonical MOP sequence (Tables 1 and 2, green color, Supplementary Table 1). An additional peptide covering positions 166 to 174 located in close proximity to the DRY motif at the end of TM3 was also identified. Nevertheless, no putative peptide corresponding to any of the non-canonical isoforms of the MOP carboxyl-terminus (Table 2, blue color) was identified by MS. Only the C-terminal

sequence [387]LENLEAETAPLP[398] that is defining the canonical MOP isoform was identified in all samples, suggesting that alternatively spliced species of MOP were either absent or present at a very low level in whole-brain extracts. However, it should be noted that a few of the putatively generated C-terminal tryptic peptides would have an amino acid length <7 that could potentially exclude them from MS detection.

In another set of experiments, we performed a series of successive immunodepletions in an attempt to enrich HA-MOP variants containing non-canonical C termini (Fig. 4). As shown in Fig. 4b, three rounds of immunodepletion with anti-HA were sufficient to remove all HA-MOP proteins from the whole brain lysate, demonstrating the high affinity and capacity of the antibody. Serial depletion of MOP receptors containing the canonical C terminus, which is selectively recognized by the

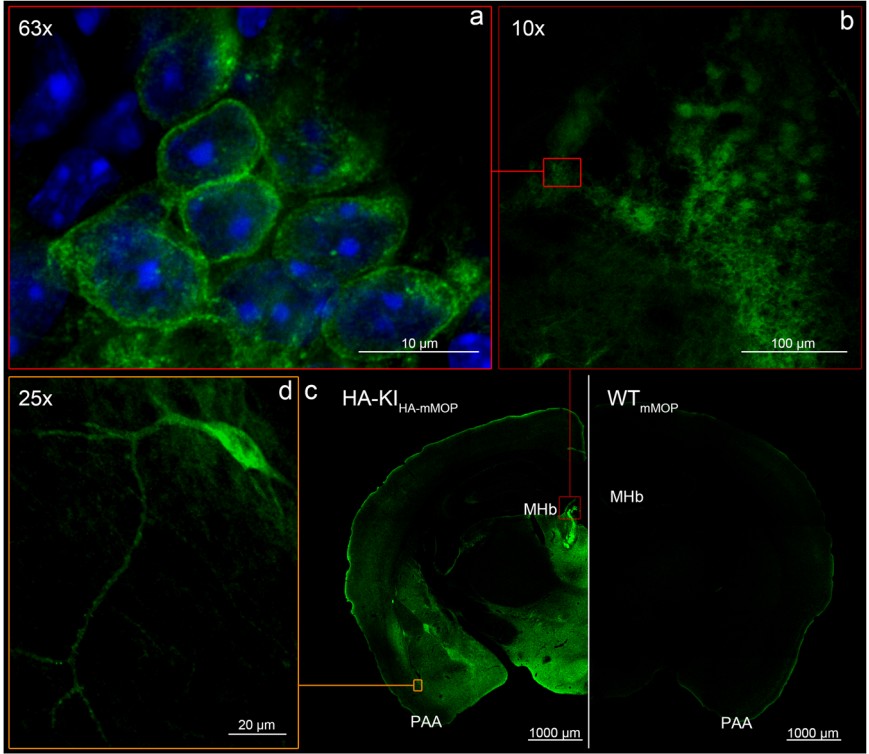

**Fig. 2 Immunostaining for HA-MOP. a**, **b** HA-MOP localization on cell bodies of habenula neurons (10× and 63× lenses). **c** 40 μm brain sections from adult HA-MOP mouse compared to wild-type. **d** Cell bodies and dendrites in the amygdala (25× lenses). cSlices were stained with Cy3-labeled (550 nm) secondary antibody and pseudo-colored afterward. Scale bars as indicated in each picture. MHb medial habenula, PAA piriform-amygdalar area.

rabbit monoclonal antibody UMB-3, removed most of HA-MOP after four rounds of precipitation (Fig. 4c). However, a very low level of HA-MOP immunopositive material remained in the lysate that could not be further depleted by additional rounds of immunoprecipitation (Fig. 4c, upper panel, precipitation rounds 6-8). A calibration standard using a serial dilution of total UMB-3 precipitated HA-MOP protein indicated that the remaining non-precipitable material corresponds to ~2% of total HA-MOP (Fig. 4c, upper and lower panel). These experiments indicate that virtually all μ-opioid receptors in brain lysates from HA-MOP knockin mice exhibit the canonical amino acid sequence.

**MOP1D splice variant**. MOP1D, also designated isoform-9, is one of the most established splice variants with several publications over the last years. A physiological function has been postulated as a receptor that specifically mediates morphine-associated itch[11,16]. Given that MOP1D is one of the isoforms that are difficult to detect by MS because of small predicted tryptic peptides at the C terminus, we generated a MOP1D-selective antibody. To address the question of whether MOP1D is expressed in the brain, we compared homogenized brain lysates from HA-MOP mice to HEK293 cells, which stably express HA-MOP1D in western blots after immunoprecipitation with anti-HA antibody beads (Fig. 5a). MOP1D was readily detected in transfected HEK293 cells (Fig. 5b, left panel) but was absent from mouse brain lysates. Conversely, canonical HA-MOP could be detected in brain lysates but not in MOP1D-expressing cells using UMB-3 that specifically detects the carboxyl-terminal $^{387}$LEN-LEAETAPLP$^{398}$ motif, which is part of the canonical MOP sequence (Fig. 5b, middle panel). Both isoforms were readily detected using the anti-HA antibody (Fig. 5b, right panel). MOP1D immunohistochemistry also failed to detect any MOP1D splice variant in the brain, spinal cord, or DRGs.

## Discussion

For all members of the opioid receptor family, transgenic mouse models have been created, in which relatively large fluorescent proteins were fused to the carboxyl terminus of the receptor[15,17–19]. While these models have proven useful to visualize the cellular receptor expression, all these models inherently harbor the risk of interference of the bulky carboxyl-terminal tag with post-translational modifications such as phosphorylation or essential protein-protein interactions, including GRKs (G protein-coupled receptor kinases), arrestins, and G proteins. One study used a conventional transgenic approach to create an N-terminally FLAG-tagged MOP mouse model and analyzed receptor desensitization and trafficking in locus coeruleus (LC) neurons[20]. However, the transgene used a tyrosine hydroxylase promoter (in order to drive expression in LC) whereas our approach is based on the targeted insertion of the tag into the endogenous mOPRM1 gene, which should preserve native expression patterns. This strategy is in line with the recently published FLAG-tagged DOP[21] and HA-tagged DOP[22] receptor mouse models that used targeted recombination. In an effort to solve this problem, we have developed a novel mouse model by fusing the HA-epitope tag sequence to the amino terminus of the MOP. Herein, we provide evidence that this approach offers a powerful means for highly efficient immuno-detection of MOP in native tissues. The HA-tag greatly facilitated immunoisolation of MOP and MS analyses confirmed post-translational modifications in vivo. Furthermore, we demonstrate agonist-dependent phosphorylation of native MOP proteins in neuronal tissue, harnessing the enhanced detection sensitivity gained by the transgenic approach.

Insertion of the N-terminal HA-tag into MOP does not impact the binding affinity of the receptor but significantly reduces HA-MOP expression levels in most brain regions. This also results in a reduced antinociceptive response to morphine in HA-MOP

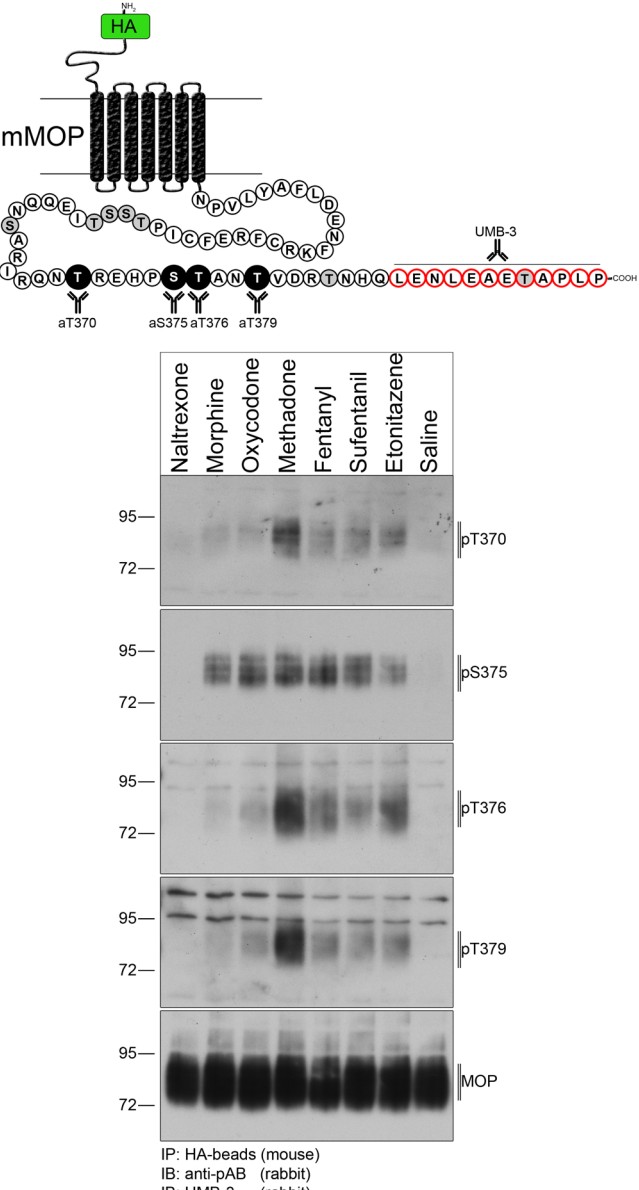

**Fig. 3 MOP phosphorylation pattern in vivo.** HA-MOP mice were either treated with different MOP agonists, naltrexone, or saline. Western blots of brain lysates immunoblotted with anti-pThr370 (first panel), anti-pS375 (second panel), anti-pThr376 (third panel), or anti-pThr379 (fourth panel) antibodies. Blots were stripped and reprobed with a UMB-3 antibody to control for equal loading. Positions of molecular mass markers are indicated on the left (in kDa). Blots are representative of four independent experiments.

knockin mice, which is similar to that observed in heterozygous MOP knockout mice[23].

Agonist-dependent multi-side phosphorylation of HA-MOP in vivo confirmed earlier experiments and recent publications on MOP phosphorylation and downstream signaling[24,25], while also demonstrating the functionality of the C-terminal part of the receptor. Previous studies using UMB-3 immunoprecipitation had identified agonist-dependent phosphorylation at T370 and S375[26], while the HA-MOP knockin mouse model showed additional phosphorylation at T376 by high-efficacy agonists, analogous to prior in vitro experiments. Therefore, this novel mouse model will greatly facilitate investigations into in vivo receptor modifications as a function of high or low efficacy

agonists, duration of opioid exposure, and their possible correlation with aversive opioid effects such as tolerance or respiratory depression.

While earlier studies already proved the expression of MOP mRNA at different prenatal stages of the mouse brain[27], with this new mouse model we were able to stain the expressed HA-tagged-protein in the CPu of the embryonic brain, starting at E16 and proceeding until P10, when there is an almost fully developed pattern of MOP in the striatum, similar to the expression that can be observed in adult mice.

Almost since the initial cloning of the opioid receptors in the early 90s, reports on alternative splice variants have raised hopes that such receptor isoforms may help explain particular physiological effects of opioid drugs. However, almost none of these numerous studies addressed the fundamental questions of quantitative significance and actual expression at the protein level of the alternative transcripts. A few studies used polyclonal antisera generated against predicted alternative MOP C termini[3,11], but none of the antisera was rigorously validated in knockout or specific transgenic mice.

We addressed the open debate about the existence of MOP isoforms containing alternative C termini from two different angles: (1) by amino acid sequencing via LC–MS/MS analysis of precipitated HA-MOP, and (2) by immunodepletion of MOP containing the canonical C terminus in order to enrich non-canonical isoforms. Both approaches failed to produce conclusive evidence for significant quantities of MOP isoforms with alternative C termini. The immunodepletion experiments indicate that non-canonical MOP receptor proteins may constitute—at best—<2% of total MOP receptors in the mouse brain and that alternatively spliced MOP transcripts may represent heteronuclear mRNA that is either not or poorly translated. Such a low abundance of MOP with non-canonical C termini is in line with the non-detection of any corresponding peptides in the LC–MS/MS analysis. However, it should be noted that the predicted tryptic hexapeptide of the postulated MOP1D C terminus might elude MS detection, because of its small size. Therefore in a third approach, we specifically investigated protein levels of the postulated MOP1D isoform by using a novel anti-MOP1D antibody. Again, these experiments failed to demonstrate any significant presence of MOP1D protein in total anti-HA-MOP precipitates. This combination of experimental approaches strongly suggests that predicted MOP isoforms with alternatively spliced C termini either do not exist in mouse brain or are present at such low levels that put their physiological significance into question.

In summary, the HA-MOP transgenic mouse model represents a valuable tool to study protein expression and activity-dependent modifications of MOP in vivo by greatly increasing the sensitivity of protein detection. MOP isoforms with alternative C termini may not exist at the protein level or are present at insignificant levels. It is therefore questionable, how such a small amount of alternative receptor protein isoforms may influence the overall functionality of MOP signaling in the CNS or other organs with even lower MOP expression levels.

## Material and methods

**Animals.** Knock-in mice expressing HA-MOP (Oprm1[em1Shlz], MGI:6117675) were generated by Applied StemCell (Menlo Park, USA), using CRISPR/Cas9-mediated targeted recombination. Mice were genotyped by PCR from genomic tail-biopsy DNA using the following primers: 5′-TACCCATACGATGTTCCA GATTACGCT-3′ (mOprm1F) and 5′-GGAACTAGGTATTCA GAACATGCCTTACCTTAC-3′ (mOprm1R), followed by RsaI restriction to detect the presence of the HA-tagged mOprm1 gene (Supplementary Fig. 1). All HA-MOP mice were backcrossed to

**Table 1 MS analysis of agonist-induced phosphorylation sites in the MOP carboxy-terminus.**

| Peptide sequence | Peptide position | MH + [Da] | # Missed cleavages | Best Mascot ion score | | | |
| | Phosphosite position | | | Best phosphoRS site probability (%) | | | |
| | | | | Saline | Morphine | Fentanyl | Etonitazene |
|---|---|---|---|---|---|---|---|
| YIAVCHPVK | 166-174 | 1086.5765 | 0 | 41 | 43 | 45 | 47 |
| EFCIPTSSTIEQQNSAR | 349-365 | 1967.9127 | 0 | 48 | 83 | 74 | 75 |
| EFCIPTSSTIEQQN**S**AR | 349-365 | 2047.8791 | 0 | | | 60 | 53 |
| | **S363** | | | | | **100** | **100** |
| IRQN**T**REHPSTANTVDR | 366-382 | 2074.9778 | 2 | | | 44 | 43 |
| | **T370** | | | | | **100** | **100** |
| IRQN**T**REHP**S**TANTVDR | 366-382 | 2154.9441 | 2 | | | 22 | 38 |
| | **T370; S375** | | | | | **99.99; 49.79** | **100; 95.64** |
| IRQN**T**REHP**S**TANTVDR | 366-382 | 2154.9441 | 2 | | 20 | 22 | 22 |
| | **T370; T376** | | | | **98.78; 89.93** | **99.99; 92.7** | **100; 92.96** |
| IRQN**T**REHP**ST**ANTVDR | 366-382 | 2234.9105 | 2 | | | | 25 |
| | **T370; S375; T376** | | | | | | **100; 99.99; 99.91** |
| QN**T**REHPSTANTVDR | 368-382 | 1805.7926 | 1 | | 29 | 21 | 42 |
| | **T370** | | | | **100** | **100** | **100** |
| QN**T**REHP**S**TANTVDR | 368-382 | 1885.7590 | 1 | | 14 | 13 | 14 |
| | **T370; S375** | | | | **98.87; 99.85** | **100; 99.3** | **99.89; 99.08** |
| TNHQLENLEAETAPLP | 383-398 | 1776.8763 | 0 | 38 | 70 | 58 | 60 |

List of MOP receptor peptides and phosphopeptides identified by nanoLC–MS/MS and their corresponding Mascot ion score in each condition. Below the Mascot score, phosphoRS site probability (%) is given for each phosphorylated amino acid position (bold numbers). Positions are indicated for the canonical protein (P42866).

**Table 2 C-terminal isoforms of mouse MOP.**

```
P42866   --ALADALATSTLPFQSVNYLMGTWPFGNILCKIVISIDYYNMFTSIFTLCTMSVDRYIAVCHPVKALDFRTPRNAKIVNVCNWILSSAIGLPVMFMATTKYRQGSIDCTLTFSHPTWYWENLLKICVFIF 239
         ******************************************************

MOP1-HA  --CFREFCIPTSSTIEQQNGARIRQNTREHPSTANTVDRTNHQLENLEAETA-----PLP------------------------------------------------------------------------ 407
MOP1     --CFREFCIPTSSTIEQQNGARIRQNTREHPSTANTVDRTNHQLENLEAETA-----PLP------------------------------------------------------------------------ 398
MOP1A    --CFREFCIPTSSTIEQQNGARIRQNTREHPSTANTVDRTNHQVCAF---------------------------------------------------------------------------------------- 390
MOP1B1   --CFREFCIPTSSTIEQQNGARIRQNTREHPSTANTVDRTNHQKIDLF--------------------------------------------------------------------------------------- 391
MOP1B2   --CFREFCIPTSSTIEQQNGARIRQNTREHPSTANTVDRTNHQKLLMWR------A-MPTF-------KRH----LAIMLSLDN------------------------------------------------- 409
MOP1B3   --CFREFCIPTSSTIEQQNGARIRQNTREHPSTANTVDRTNHQTSLTLQ-------------------------------------------------------------------------------------- 392
MOP1B4   --CFREFCIPTSSTIEQQNGARIRQNTREHPSTANTVDRTNHQAHQKPQ------E-CLKC-------RCLSLTILVICLHFQHQQFFIMIKKNVS---------------------------------------- 425
MOP1B5   --CFREFCIPTSSTIEQQNGARIRQNTREHPSTANTVDRTNHQCV------------------------------------------------------------------------------------------ 388
MOP1C    --CFREFCIPTSSTIEQQNGARIRQNTREHPSTANTVDRTNHQPTLAVSVAQIFTGYPSPTHVEKPCKSCMD---------------------------------RGMRNLLPDDGPRQESGEGQLGR 438
MOP1D    --CFREFCIPTSSTIEQQNGARIRQNTREHPSTANTVDRTNHQRNEEPS------S----------------------------------------------------------------------------- 393
MOP1E    --CFREFCIPTSSTIEQQNGARIRQNTREHPSTANTVDRTNHQKKKLDSQ----RG-CVQH-------PV---------------------------------------------------------------- 401
MOP1F    --CFREFCIPTSSTIEQQNGARIRQNTREHPSTANTVDRTNHQAPCACVPGAN-RG-QTKA------SDLLDLELETVGSHQADA------ETNPGPYEGSKCAEPLAISLV-PLY-------------- 444
MOP1O    --CFREFCIPTSSTIEQQNGARIRQNTREHPSTANTVDRTNHQPTLAVSVAQIFTGYPSPTHVEKPCKSCMD---------------------------------------------------------- 416
MOP1P    --CFREFCIPTSSTIEQQNGARIRQNTREHPSTANTVDRTNHQIMKFEAIYPK-LSFKSWA------LKYFTFIREKKRNTKAGALPTC-HAGS---PS---QAHRGVAAWLL-PLRHMGPSYPS------- 453
MOP1U    --CFREFCIPTSSTIEQQNGARIRQNTREHPSTANTVDRTNHQPTLAVSVAQIFTGYPSPTHVEKPCKSCMDSVDCYNRKQQTGSLRKNKKKKKRRKNKQNILEAGISRGMRNLLPDDGPRQESGEGQLGR 474
MOP1V    --CFREFCIPTSSTIEQQNGARIRQNTREHPSTANTVDRTNHQKKQEKTKTKSAWEIWEQKEHTLLLGETHLTIQ------H----LS----------------------------------------- 420
MOP1W    --CFREFCIPTSSTIEQQNGARIRQNTREHPSTANTVDRTNHQLAFGCCNEHHDQR---------------------------------------------------------------------------- 399
         **********************************************
```

Alignment of MOP splice variant sequences. UniProtKB/Swiss-Prot identifiers are given in Supplementary Table S1. Isoforms predicted not to contain the HA-tag have been omitted. Only portions of the sequence containing MS-identified peptides or putative splice variant alternative peptides are shown. *, fully conserved part of the sequence. In green, peptides are identified by nanoLC–MS/MS. In blue, other putative splice variant peptides that could have been detected by MS. Highlighted in gray, trypsin cleavage sites. Highlighted in red, phosphorylated residues identified by nanoLC–MS/MS.

JAX™ C57Bl/6J obtained from Charles River Laboratories (DE) which were also used for breeding of mutant strains and served as controls in all experiments. Animals were housed 2–5 per cage under a 12-hr light-dark cycle with ad libitum access to food and water. All animal experiments were performed in accordance with relevant guidelines and regulations, were approved by Thuringian state authorities, and complied with European Commission regulations for the care and use of laboratory animals. Our study is reported in accordance with the ARRIVE (Animal Research: Reporting In Vivo Experiments) guidelines[28]. In all experiments, male and female mice aged 8–30 weeks between 25 and 35 g body weight were used.

**Drugs and routes of administration.** All drugs were freshly prepared prior to use and were injected subcutaneously in lightly restrained, unanaesthetized mice at a volume of 10 $\mu l^{-1}$ body-weight. Drugs were diluted in 0.9% (W/V) saline for injections.

Drugs were obtained and used as follows: morphine sulfate (30 mg kg$^{-1}$ for 30 min; Hameln Inc., Hameln, Germany), oxy-codone hydrochloride (15 mg kg$^{-1}$ for 30 min; Mundipharma GmbH, Limburg, Germany), levomethadone hydrochloride (15 mg kg$^{-1}$ for 30 min; Sanofi-Aventis, Frankfurt, Germany), sufentanil (30 mg kg$^{-1}$ for 15 min; Hameln Inc., Hameln, Germany), fentanyl citrate (0.3 mg kg$^{-1}$ for 15 min; Rotexmedica, Trittau, Germany), etonitazene (30 mg kg$^{-1}$ for 30 min; Sigma-Aldrich, Munich, Germany) and naltrexone hydrochloride (10 mg kg$^{-1}$ overnight; Neuraxpharm, Langenfeld, Germany).

**Reagents and antibodies.** Pierce™ Anti-HA Magnetic Beads were obtained from Thermo Fisher Scientific (Schwerte, Germany), while the phosphorylation-independent antibodies were obtained as follows: rabbit monoclonal anti-HA antibody (Cell Signaling, Frankfurt, Germany), and anti-MOP antibody {UMB-3} (Epi-tomics, Burlingame, CA); used as previously described[29]. The

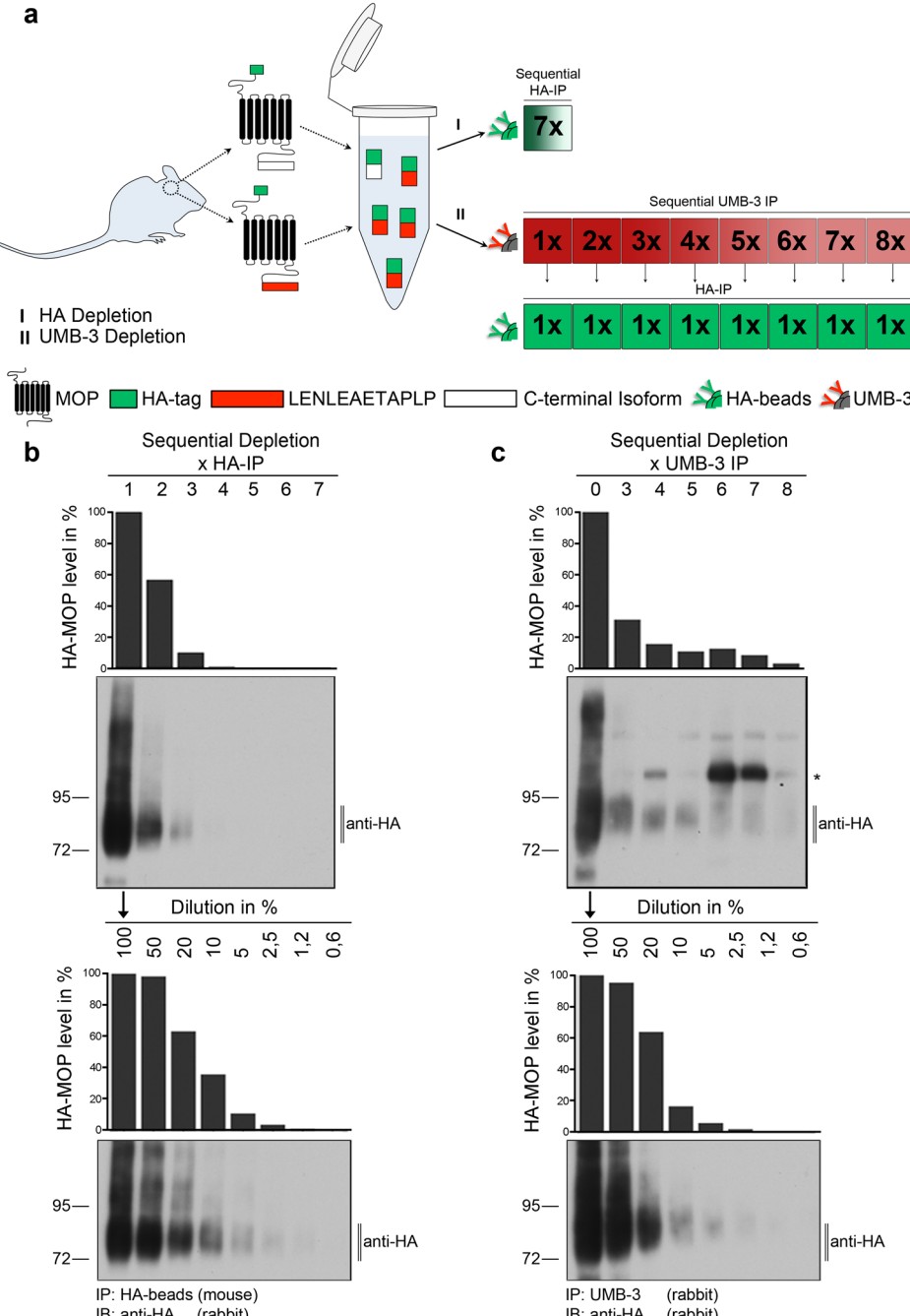

**Fig. 4 Canonical HA-MOP immunodepletion experiment. a** Schematic representation of the experimental procedure. **b**, **c** (Upper panel) Depletion of HA-MOP by several immunoprecipitation steps with either. **b** HA-tagged magnetic beads or (**c**) UMB-3-protein-A beads-conjugate. **b**, **c** (Lower panel) Dilution series of a single HA-IP sample. Results were quantified using ImageJ and Prism software. Representative results from three independent experiments are shown. The position of molecular mass markers is indicated on the left (in kDa). *Non-specific antibody band in **c**. Blots are representative of eight independent experiments.

rabbit polyclonal phosphosite-specific μ-opioid receptor antibodies anti-pT370 (7TM0319B), anti-pT376 (7TM0319D) and anti-pT379 (7TM0319E) were obtained from 7TM Antibodies (Jena, Germany)[24,30,31]. The polyclonal phosphosite-specific anti-pS375 was obtained from Cell Signaling (Frankfurt, Germany). The polyclonal rabbit phosphorylation-independent-antibody for MOP1D was generated by S.S. the founder and scientific advisor of 7TM Antibodies GmbH, Jena, Germany against the alternative C-terminal splice sequence NHQRNEEPSS. This sequence corresponds to amino acids 384-393 of the postulated MOP1D mouse receptor. The antibodies were affinity-purified against

their immunizing peptide using the SulfoLink kit (Thermo Scientific, Rockford, IL). In addition, the following commercially available secondary antibodies were used: polyclonal donkey anti-rabbit IgG Cy3 (Dianova, Hamburg, Germany) and goat anti-rabbit IgG, HRP-linked antibody (Cell Signaling, Frankfurt, Germany).

**Immunoprecipitation of HA-MOP from brain lysates.** Depending on the experiment, mice were either treated with agonists, antagonists, saline or received no treatment. Mice were

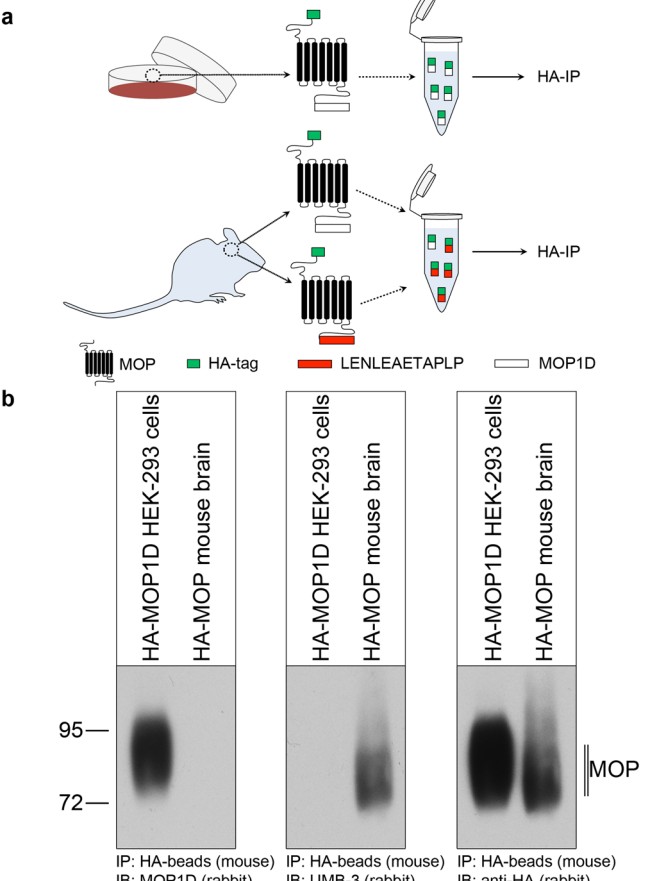

**Fig. 5 Western blot analysis of MOP1D expression in vivo. a** Schematic representation of the experimental procedure **b** HEK293 cells stably expressing HA-MOP1D, as well as brain lysates from HA-MOP mice, were immunoblotted with anti-MOP1D antibody (left panel), UMB-3 (central panel), or anti-HA antibody (right panel). Representative data from one of four independent experiments are shown. Positions of molecular mass markers are indicated on the left (in kDa). Blots are representative of four independent experiments.

anesthetized with isoflurane, killed by cervical dislocation, and brains were quickly dissected, excluding the cerebellum. Brain samples were immediately frozen in liquid nitrogen. Brains were transferred to ice-cold detergent buffer (50 mM Tris-HCl, pH 7.4, 150 mM NaCl, 5 mM EDTA, 1% Nonidet P-40, 0.5% sodium deoxycholate, 0.1% sodium dodecyl sulfate (SDS), containing protease and phosphatase inhibitors), homogenized, and centrifuged at $14,000 \times g$ for 30 min at 4 °C. The supernatant was then precipitated with HA-tagged magnetic beads (ThermoFisher Scientific, Germany) for 60 min at 4 °C. Afterward, the receptor beads conjugates were separated from the supernatant using a special magnetic device (DynaMag$^{TM}$−2, life technologies) and washed three times. Proteins were eluted from the beads with SDS sample buffer for 25 min at 43 °C and then resolved on 8% SDS–polyacrylamide gels. After electroblotting, membranes were incubated with anti-pT370, anti-pS375, anti-pT376, or anti-pT379 antibody, followed by detection using a chemiluminescence detection system. Blots were subsequently stripped and incubated again with the phosphorylation-independent antibodies anti-HA and UMB-3 to confirm equal loading of the gels. Films exposed in the linear range were then densitized using ImageJ 1.37v.

The same procedure was used for the MOP1D detection experiments. Membranes were incubated with either anti-MOP1D or UMB-3 antibodies, followed by detection using a chemiluminescence detection system. Blots were subsequently stripped and incubated again with the phosphorylation-independent antibody anti-HA to confirm equal loading of the gels.

**Immunodepletion of canonical HA-MOP.** In order to enrich MOP variants that are alternatively spliced at the C terminus, we employed immunodepletion experiments. Brains were dissected from untreated HA-MOP mice, homogenized as described above and supernatants were pooled. Using the well-characterized antibody UMB-3, receptor proteins containing the canonical carboxyl-terminal $^{387}$LENLEAETAPLP$^{398}$ motif were successively removed by immunoprecipitation using protein-A-agarose beads. In theory, only MOP variants with non-canonical C termini should thus remain in the lysate but should be detectable using their N-terminal HA-tag. A total of seven successive rounds of immunoprecipitation were performed and an aliquot was removed after each step. From each aliquot, remaining HA-MOP was precipitated using HA-beads as described and captured proteins were analyzed by western blot probing for HA-epitopes. The quantitative capacity of anti-HA to precipitate HA-MOP was evaluated using the same immunodepletion strategy. Brain lysates were treated as described above and successively immunoprecipitated using HA-beads for seven rounds. Aliquots from each step were analyzed by western blot. Proteins were then loaded in a dilution series from 100 to 0.6% on 8% SDS–polyacrylamide gel. Staining intensities from these blots were used as a calibration standard to evaluate remaining HA-tagged proteins in the immunodepletion experiments.

**Cell culture and transfection.** HEK293 (human embryonic kidney 293 cells) cells were obtained from the German Resource Centre for Biological Material (DSMZ, Braunschweig, Germany) and grown in Dulbecco's modified Eagle´s medium supplemented with 10% fetal calf serum in a humidified atmosphere containing 5% $CO_2$. Cells were transfected with plasmid encoding murine HA-tagged MOP or MOP1D using Lipofectamine according to the instructions of the manufacturer (Invitrogen, Carlsbad, CA). Stable transfectants were selected in the presence of 1 µg ml$^{-1}$ puromycin for HA-MOP or G-418 500 µg ml$^{-1}$ for HA-MOP1D. HEK293 cells stably expressing MOP were characterized using radioligand-binding assays, western blot analysis, immunocytochemistry, and cAMP assays as described previously[6]. For western blot analysis, cells were seeded onto poly-L-lysine-coated 60 mm dishes and grown to 90% confluence. Cells were lysed in RIPA buffer (50 mM Tris-HCl, pH 7.4, 150 mM NaCl, 5 mM EDTA, 1% Nonidet P-40, 0.5% sodium deoxycholate, 0.1% SDS) containing protease and phosphatase inhibitors (Complete mini and PhosSTOP; Roche Diagnostics, Mannheim, Germany). Pierce$^{TM}$ HA-epitope tag Antibodies (Thermo Scientific, Rockford, USA) were used to enrich HA-tagged MOP. To elute proteins from the beads, the samples were incubated in an SDS sample buffer for 25 min at 43 °C. Supernatants were separated from the beads, loaded on 8% SDS–polyacrylamide gels, and immunoblotted onto nitrocellulose afterward. After blocking, membranes were incubated with either MOP1D antibody or UMB-3 antibody at 4 °C overnight. On the next day, membranes were incubated with peroxidase-conjugated secondary antibody followed by detection using a chemiluminescence system (90 mM p-coumaric-acid, 250 mM luminol, 30% hydrogen peroxide). Afterward, blots were stripped and reprobed with an anti-HA

antibody to confirm equal loading of the gel. Protein bands on western blots were exposed to X-ray films.

**Immunohistochemistry**. Mice were anesthetized with isoflurane and transcardially perfused with Tyrode's solution followed by Zamboni's fixative (4% paraformaldehyde and 0.2% picric acid in 0.1 M phosphate buffer, pH 7.4). Brains were rapidly dissected and postfixed in the same solution for 2 hours. Then the tissue was cryoprotected by immersion in 10% sucrose, followed by 30% sucrose for 48 h at 4 °C before sectioning using a freezing microtome. Free-floating sections (40 μm) were washed multiple times, blocked, and incubated with anti-HA antibody (Cell Signaling, Frankfurt, Germany) overnight. On the following day, Cy3-conjugated anti-rabbit antibody (Dianova, Hamburg, Germany) was used for detection. Cy3 was imaged with excitation at 568 nm using a Zeiss LSM510 META laser scanning confocal microscope or a Zeiss LSM 900 Airyscan 2 equipped with ZEN software for image analysis.

**In-gel tryptic digestion, nanoLC–MS/MS analysis, and database searches**. For mass spectrometry (MS) analysis, HA-immunoprecipitated samples were reduced for 30 min at 37 °C by adding 1×SDS sample buffer containing 30 mM DTT, and then alkylated in 90 mM iodoacetamide for 30 min in the dark at room temperature. A volume of 60 μl of reduced/alkylated protein samples was separated by SDS-PAGE on 10% polyacrylamide gels followed by gel staining with colloidal Coomassie blue. A band was excised at the level of the smear detected by western blot and subjected to in-gel tryptic digestion using modified porcine trypsin (Promega, France) at 20 ng μl$^{-1}$. Tryptic peptides were extracted and analyzed in triplicate by online nanoLC using an Ultimate 3000 system (Dionex, Amsterdam, The Netherlands) coupled to an ETD-enabled LTQ Orbitrap Velos mass spectrometer (ThermoFisher Scientific, Bremen, Germany) as previously described[26,32]. Survey scan MS was performed in the Orbitrap over a 300–2000 $m/z$ mass range with resolution set to a value of 60,000 at $m/z$ 400. The 20 most intense ions per survey scan were selected for subsequent CID (collision-induced dissociation)/ETD (electron transfer dissociation) fragmentation, and the resulting fragments were analyzed in the linear trap (LTQ). The settings for the data-dependent decision tree-based CID/ETD method were as follows: ETD was performed instead of CID if charge state was 3 and $m/z$ < 650, or if the charge state was 4 and the $m/z$ < 900, or if the charge state was 5 and the $m/z$ < 950. The ETD was performed for all precursor ions with charge states > 5. The normalized collision energy was set to 35% for CID. The reaction time was set to 100 ms and supplemental activation was enabled for ETD. Dynamic exclusion was employed within 30 s to prevent repetitive selection of the same peptide. For internal calibration, the 445.120025 ions was used as lock mass. Three to four technical replicates were performed for each condition. All raw MS files were processed with Proteome Discoverer software (version 2.1, ThermoFisher Scientific) for database search with the Mascot search engine (version 2.6.0, Matrix Science, London, UK) combined with the Percolator algorithm (version 2.05) for PSM search optimization and the phosphoRS algorithm (version 3.1,[33]) for phosphorylation site localization. For both fragmentation techniques, the parameters set for the creation of the peak lists were: parent ions in the mass range 300–5000 Da and no grouping of MS/MS scans. The non-fragmented filter was used to simplify ETD spectra with the following settings: the precursor peak was removed within a 4 Da window, charged reduced precursors were removed within a 2 Da window, and neutral losses from charged reduced precursors were removed within a 2 Da window (the maximum neutral loss mass was set to 120 Da). Peak lists were searched against SwissProt database with taxonomy *Mus musculus* (16761 sequences) implemented with the mouse HA-tagged MOP receptor sequence and the sequences of 19 predicted MOP isoforms produced by alternative splicing already described in the UniProt entry. Enzyme specificity was set to trypsin/P and a maximum of three missed cleavages were allowed. Carbamido-methylation of cysteine was set as fixed modification whereas oxidation of methionine and phosphorylation of serine, threonine, and tyrosine were set as variable modifications. Mass tolerances in MS and MS/MS were set to 10 ppm and 0.6 Da, respectively. Mascot results were validated by the target-decoy approach using a reverse database of the same size. The Percolator algorithm was used to calculate a q-value for each peptide-spectrum match (PSM), peptides and PSM were validated based on Percolator q-values at a False Discovery Rate (FDR) set to 5%. Then, peptide identifications were grouped into proteins according to the law of parsimony and filtered to 5% FDR.

**Statistics and reproducibility**. Western blot data were analyzed using ImageJ and all calculations were performed using Graph-Pad Prism software (GraphPad Software, Inc., San Diego, CA). Wherever appropriate, data were analyzed using unpaired *t*-tests with significance set at $P < 0.05$. Responses to cumulative doses of morphine were analyzed by two-way ANOVA, followed by Bonferroni's post hoc test[34–38].

**Reporting summary**. Further information on research design is available in the Nature Research Reporting Summary linked to this article.

### Data availability

The authors declare that all data supporting the findings of this study are available within the paper and its supplementary information files. All Mass spectrometry data are available to ProteomeXchange via the PRIDE database. The data that support the findings of this study are available from the authors upon reasonable request.

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

## Acknowledgements

We thank Helga Bechmann, Heike Stadtler, and Olga Trovato for excellent technical assistance. This work was supported by the Else Kröner Fresenius Stiftung (2019_A68), Interdisciplinary Center for Clinical Research Jena (AMSP 03) to A.K. and Deutsche Forschungsgemeinschaft grants SFB/TR166-TPC5, SCHU924/15-1 and SCHU924/18-1 to S.S. and in part by the Région Occitanie, European funds (Fonds Européen de Développement Régional, FEDER), Toulouse Métropole, the French Ministry of Research with the Investissement d´Avenir Infrastrucutures Nationales en Biologie et Santé (ProFI, Proteomics French Infrastructure project, ANR-10-INBS-08) and the Austrian Science Fund (FWF: I2463-B21) to M.S.

## Author contributions

S.S. initiated the project and designed all experiments with A.K. A.K. performed in vivo phosphorylation and behavior studies. S.F. performed all in vitro and in vivo phosphorylation studies. Mass spectrometry experiments were performed by L.M., C.M. and C.F. and analyzed under the supervision of O.B.-S. Autoradiography was performed by F.E. and A.B. M.S. performed the [³H]DAMGO saturation binding assay and [³⁵S] GTPγS binding assay. The manuscript was written and revised by S.S., A.K., S.F. and R.K.R. with input from other authors.

## Funding

## Competing interests

S.S. is the founder and scientific advisor of 7TM Antibodies GmbH, Jena, Germany, and declares no competing non-financial interests but competing financial interests. The remaining authors declare no competing interests: details are available in the online version of the paper.
