## [Peer Review File · Communications Biology]

Reviewers' comments:

Reviewer #1 (Remarks to the Author):

This manuscript characterizes a novel mouse line in which the endogenous mu opioid receptor (MOR) is replaced with a HA-tagged MOR. This tag is a significant advancement to the MOR-cherry reporter mice, as the HA tag is less obtrusive, still allows for imaging, and facilitates mass spec experiments. The groups uses this novel mouse line to determine the existence of MOR splice variants. There is controversy surrounding the existence of MOR splice variants and their putative physiological roles; and the utilization of the HA-MOR mice provides support for the argument that splice variants of MOR do not exist in mice. Overall, the experiments are straightforward and clear, however, I do have a few concerns:

1) The statistical analysis used in this study should be more clearly stated at the end of the Methods.

2) The HA-MOR mouse is a useful tool that could be of great benefit to the scientific community. For this reason, a better characterization of the possible imaging applications of these mice should be done. The fact that HA-MOR can be detected with immunohistochemistry using an anti-HA antibody is already a step better than the recently developed HA-DOR mice. What does HA-MOR look like at higher magnification in slices? Is the cellular resolution sufficient for co-expression or in vivo trafficking studies? It would be a missed opportunity to not perform these types of studies.

3) In Figure 2 it would be nice to include similar blots from a WT mouse. This group has published this characterization previously but it would be important to include in this paper as an internal control/replication as well as to allow direct comparison with the HA-MOR mice.

4) The authors develop a MOP1D antibody and show MOP1D detection in transfected HEK293 cells but none in HA-MOP mice. Does this antibody detect anything in WT mice?

Reviewer #2 (Remarks to the Author):

In their manuscript, Fritzwanker and colleagues report the generation of a new knockin mouse expressing an N-terminal HA-tagged MOP. This novel mouse was used here to study 1) the agonist-specific phosphorylation of the C-term LENLEAETAPLP motif and 2) the potential existence of MOP splice variants. The strenght of the study clearly is the fact that these experiments were conducted in vivo. As opposed to many other proteins, the study of GPCRs in vivo is challenging as these receptors are usually expressed at very low levels (fmol/mg of proteins, at the very best) and their 7 transmembrane domain conformation restrict their solubility and the possibilities to generate specfici antibodies (very few example of specific and highly reliable antibodies raised against GPCRs have been reported to date).

Although this mouse model has a huge potential for the field, there are a number of important concerns that needs to be addressed before this manuscript become acceptable for publication.

1- Since this is a novel mouse model that has never been published before, more details are needed. How and where the HA tag has been inserted? Are these mice behave normally? Do they respond to morphine? How they compare to WT animals in terms of MOP expression in the brain (Bmax and EC50)? This is crucial to shown that they are equivalent to normal animal expressing the endogenous receptor.

2- I have concerns with the mass spec analysis and the conclusions raised by the authors (i.e. absence of splice variants). It is always very challenging to shown an absence of something of to provide sufficient eviudence for a negative result. In this study, one should admit that the fact that peptides are not found by mass spec does not prove that a protein is not present. GPCRs, as acknowledged by the authors, often generate (too) small tryptic peptides (less prone to be detected in mass spec). Additionally, for other reasons (ex. lack of solubility) some predicted sequence cannot be found as they are not accessible to trypsin. How the authors can make sure

they have not missed something because of such phenomenon?

3- GPCRs usually migrate at various apparent MW on SDS gel. This is very observed by the authors in Figure 1c. It is not clear to me why the authors have elected to cut the gel at the predicted MW of HA-MOP when brain HA-MOP apparently distributes (as per Fig 1c) anywhere between 50 and 150 kDa. It may very well happen (at least, it cannot be excluded) that variants are modified or assemble in a SDS-resistant complex to migrate at different MW.

4- In addition to the concern raised in 3, In-gel trypsin digestion may restrict the access the some cleavage sites. A recent publication from Degrandmaison et al (PNAS 2020) rather used an on-beads digestion approach which I think could be more appropriate here as well as it would not exclude any forms/isoforms migrating to a different MW. Admittedly, such an approach reduce the purity of the sample, but it also gives an opportunity to avoid drawbacks of the in-gel digestion approach (expecially for GPCRs).

5- Previous publications (at least 3) report the generation of knockin mice for opioid receptors (one for MOP and two for DOP). Theses should be acknowledged. Most importantly, this report is not the first to show the use of epitope-tagged OP receptors to study the expression and mass spec in vivo.

Degrandmaison et al. PNAS 2020 (Flag-DOP KI mice - mass spec)

Su et al. Sci Rep (2017) (HA-DOP - immuno detection)

Arttamangkul et al. Mol Pharmacol (2008) (Flag-MOP - immuno detection)

6- It is the opinion of this reviewer that authors should tune down their conclusion on the existence of MOP splice variants. I do not think that this report convincingly show that such variants do not exist, nor that if they do, they are expressed at a level that possibly has no physiological significance. In fact, even a very low level of expression in a discrete population of cells might play a significant physiological role. In the whole brain, the total amount might seem too low to have any role, but if concentrated in a specific area/nucleus, who knows?

Minor comments:

1- I couldn't find how many separate experiments have been conducted.

2- Predicted MW for each splice variants should be provided

June 21, 2021

Revision of manuscript: COMMSBIO-20-2249

Dear Reviewers,

We are grateful to the Reviewers for critically reviewing our manuscript entitled " HA-MOP knockin mice express the canonical μ -opioid receptor but lack detectable splice variants" We have now completed the revision of our manuscript, taking all of the Reviewers' comments and criticisms into account. They prompted revision of the text and figures and additional experiments, which has strengthened our manuscript. Please see the detailed responses below (blue text). New manuscript text is in red.

Reviewer 1 comments

This manuscript characterizes a novel mouse line in which the endogenous mu opioid receptor (MOR) is replaced with a HA-tagged MOR. This tag is a significant advancement to the MOR-cherry reporter mice, as the HA tag is less obtrusive, still allows for imaging, and facilitates mass spec experiments. The groups uses this novel mouse line to determine the existence of MOR splice variants. There is controversy surrounding the existence of MOR splice variants and their putative physiological roles; and the utilization of the HA-MOR mice provides support for the argument that splice variants of MOR do not exist in mice. Overall, the experiments are straightforward and clear, however, I do have a few concerns:

1) The statistical analysis used in this study should be more clearly stated at the end of the Methods. We added a description of the statistical analysis at the end of the Methods section on page 11.

2) The HA-MOR mouse is a useful tool that could be of great benefit to the scientific community. For this reason, a better characterization of the possible imaging applications of these mice should be done. The fact that HAMOR can be detected with immunohistochemistry using an anti-HA antibody is already a step better than the recently developed HA-DOR mice. What does HA-MOR look like at higher magnification in slices? Is the cellular resolution sufficient for co-expression or in vivo trafficking studies? It would be a missed opportunity to not perform these types of studies.

We thank the reviewer for the suggestion and included a new Figure 2 (page 13, 24 and 29). We can show HA-MOP at high resolution in the habenula and amygdala, located on cell bodies as well as putative dendrites.

3) In Figure 2 it would be nice to include similar blots from a WT mouse. This group has published this characterization previously but it would be important to include in this paper as an internal control/replication as well as to allow direct comparison with the HA-MOR mice.

As the reviewer correctly mentions, we have already published the full in vivo MOP S375 phosphorylation pattern after saline, morphine, fentanyl and etonitazene injection in WT mice (Doll et al., 2012; Glück et al., 2014). MOP IP from WT mice needs first the precipitation of the MOP with UMB-3 (rabbit) bound protein A beads from the brain lysates. Phosphorylation is then detected with phospho-specific antibodies from guinea pig. Since the affinity and capacity of UMB-3 and anti-HA antibodies are significantly different and the methodology for MOP immunoprecipitation is less efficient from WT brains than HA-MOP knockin brains, such an experiment would create the false

impression of low MOP abundance in WT brains compared to HA-MOP brains. We believe that our additional data from saturation radioligand binding and functional assays and quantitative autoradiography (see comment 1 of Reviewer 2) provide better evidence on the actual protein expression than using two inherently different methods for immunoprecipitation. Therefore, we have not performed the experiment that the reviewer suggests.

4) The authors develop a MOP1D antibody and show MOP1D detection in transfected HEK293 cells but none in HA-MOP mice. Does this antibody detect anything in WT mice?

Due to the inherently low expression levels of native MOP, some form of "enrichment" is always necessary before the protein can be detected in Western blots. For native MOP, UMB-3 is currently the only available antibody that can assist such an enrichment step. However, UMB-3 is specifically recognizing the canonical C-terminal sequence and thus would exclude MOP1D. To circumvent this limitation and to generate an alternative approach for MOP enrichment, we have developed the HA-MOP mouse model from which MOP receptor proteins can be efficiently enriched using anti-HA antibodies. However, we have not yet tried to use the anti-MOP1D antiserum together with protein A-beads to attempt immunoprecipitation from WT brains. It is quite unlikely that any MOP1D-like protein will be detected after we saw no specific staining in the highly enriched immunoprecipitate from HA-MOP brains, but we will keep this suggestion in mind for future experiments.

Reviewer 2 comments

In their manuscript, Fritzwanker and colleagues report the generation of a new knockin mouse expressing an N-terminal HA-tagged MOP. This novel mouse was used here to study 1) the agonist-specific phosphorylation of the C-term LENLEAETAPLP motif and 2) the potential existence of MOP splice variants. The strength of the study clearly is the fact that these experiments were conducted in vivo. As opposed to many other proteins, the study of GPCRs in vivo is challenging as these receptors are usually expressed at very low levels (fmol/mg of proteins, at the very best) and their 7 transmembrane domain conformation restrict their solubility and the possibilities to generate specific antibodies (very few example of specific and highly reliable antibodies raised against GPCRs have been reported to date).

Although this mouse model has a huge potential for the field, there are a number of important concerns that needs to be addressed before this manuscript become acceptable for publication.

1) Since this is a novel mouse model that has never been published before, more details are needed. How and where the HA tag has been inserted? Are these mice behave normally? Do they respond to morphine? How they compare to WT animals in terms of MOP expression in the brain (Bmax and EC50)? This is crucial to shown that they are equivalent to normal animal expressing the endogenous receptor.

1. We included a Supplementary Figure 1, that shows the insertion and genotyping strategy in detail. More details are also provided in the Methods (page 6) and Results (page 13) sections.

2. We added a Supplementary Figure 2 which shows the hot plate test, quantitative autoradiography of [³H]DAMGO specific binding in brain slices, as well saturation binding of [³H]DAMGO and DAMGO-stimulated [³⁵S]GTPγS binding in membranes from WT and HA-MOP mice. Additional data are presented in the Results (page 13) and the Discussion (page 17) sections.

Results: "HA-MOP knockin mice were viable and healthy and showed no differences in body weight and basal pain response (Suppl. Fig. 2d, e). Binding affinity (K_d) of [³H]DAMGO determined in

saturation binding studies using brain membranes was not changed between WT and HA-MOP mice (Suppl. Fig. 2a). However, the total numbers of receptors (B_{max}) were significantly decreased by about 50% in the brain from HA-MOP mice (Suppl. Fig. 2b). Similar results were obtained by quantitative autoradiography of [3 H]DAMGO binding in the nucleus accumbens, thalamus, hypothalamus, amygdala and somatosensory cortex (Suppl. Fig. 2c). Furthermore, DAMGO-stimulated [35 S]GTP γ S binding is significantly decreased in HA-MOP brain membranes compared to wild-type, both in potency (EC_{50}) and efficacy (% stimulation). Accordingly, HA-MOP knockin mice exhibit reduced antinociceptive responses to morphine, which are similar to heterozygous MOP knockout (MOP $^{+/-}$) mice (Suppl. Fig. 2f)."

Discussion: "Insertion of the N-terminal HA-tag into MOP does not impact binding affinity of the receptor but significantly reduces HA-MOP expression levels in most brain regions. This also results in a reduced antinociceptive response to morphine in HA-MOP knockin mice, which is similar to that observed in heterozygous MOP knockout mice³⁶."

2) I have concerns with the mass spec analysis and the conclusions raised by the authors (i.e. absence of splice variants). It is always very challenging to show an absence of something or to provide sufficient evidence for a negative result. In this study, one should admit that the fact that peptides are not found by mass spec does not prove that a protein is not present. GPCRs, as acknowledged by the authors, often generate (too) small tryptic peptides (less prone to be detected in mass spec). Additionally, for other reasons (ex. lack of solubility) some predicted sequence cannot be found as they are not accessible to trypsin. How can the authors make sure they have not missed something because of such a phenomenon?

We agree that not detecting alternative peptides does not exclude the presence of very small amounts of splice variants, which was mentioned in the manuscript page 14 "..., suggesting that alternatively spliced species ... present at low level in whole brain extract". This is why we present our results as a "lack of detectable splice variants". This is also the reason why we complemented our MS study with experiments of selective immunodepletion of the canonical MOP to assess the presence of other remaining forms. In addition, the use of a specific antibody did not allow detecting MOP1D. In this sense, we have rephrased the sentence on page 17 "Such a low abundance of MOP... with the absence of corresponding peptides in the LC-MS/MS analysis." by the following one "Such a low abundance of MOP with non-canonical C-termini is in line with the non-detection of any corresponding peptides in the LC-MS/MS analysis." which is less categorical. More specifically, we agree with the reviewer concerns about missing some tryptic peptides because of their small size (already acknowledged in the manuscript), their lack of solubility or the inaccessibility of the cleavage site. However, all the putative peptides that we were looking for come from the soluble C-terminal part of the receptor. From our previous experience with MS analysis of other GPCRs (NPFF2, NOP) the C-terminal peptides are usually well detected in every sample.

Finally, we manually checked the presence or not of the MS ions corresponding to over a hundred putative non-canonical C-terminal peptides by performing Extracted-ion chromatogram (XIC) on each putative m/z in each raw file using Xcalibur software (version 3.0.63, Thermo Fisher Scientific). The possibly observable (considering most probable charge states and possible modifications: carbamidomethylation of cysteine as fixed modification and oxidation of methionine as variable modification), and detectable (considering the 300–2000 m/z mass range used for acquiring the survey scan MS) m/z for all C-term peptides were computed using PeptideMass on ExPASy.org for

each MOP isoform. This manual search did not result in a positive identification for any of the putative peptides.

3) GPCRs usually migrate at various apparent MW on SDS gel. This is very observed by the authors in Figure 1c. It is not clear to me why the authors have elected to cut the gel at the predicted MW of HA-MOP when brain HA-MOP apparently distributes (as per Fig 1c) anywhere between 50 and 150 kDa. It may very well happen (at least, it cannot be excluded) that variants are modified or assemble in a SDS-resistant complex to migrate at different MW.

We apologize for this mistake in the description of the method. We did not use the predicted molecular weight of MOP receptors (ranging from 44.4 to 54 kDa as now indicated in the Table S1) since the glycosylated receptors migrate at higher MW. We actually cut a large band of gel at the level of the smear detected by Western-blot, between the 72 and 95 kDa MW markers. This has now been corrected in the Methods section. We agree that oligomeric forms can be detected by Western-blot with long exposure time but in SDS gels the monomeric forms should be the most abundant.

4) In addition to the concern raised in 3, In-gel trypsin digestion may restrict the access to some cleavage sites. A recent publication from Degrandmaison et al (PNAS 2020) rather used an on-beads digestion approach which I think could be more appropriate here as well as it would not exclude any forms/isoforms migrating to a different MW. Admittedly, such an approach reduces the purity of the sample, but it also gives an opportunity to avoid drawbacks of the in-gel digestion approach (especially for GPCRs).

As already mentioned above we do not think that solubility and accessibility are a concern for the C-terminal soluble peptides that we are looking for. Moreover, the immunodepletion data confirm that we are dealing with very low amounts of non-canonical MOP receptor proteins. The on-beads digestion approach described by Degrandmaison et al (PNAS 2020) seems appropriate in the context of interacting partner identification but not for the characterization of expected low abundance splice variants. Indeed, increasing the complexity of the sample by skipping the gel-based fractionation step is expected to increase the dynamic range and decrease the sensitivity of the analysis. Moreover the presence of tens of putative interacting partners in a non-denatured sample is very likely to prevent trypsin access to its cleavage sites.

5) Previous publications (at least 3) report the generation of knockin mice for opioid receptors (one for MOP and two for DOP). These should be acknowledged. Most importantly, this report is not the first to show the use of epitope-tagged OP receptors to study the expression and mass spec in vivo. Degrandmaison et al. PNAS 2020 (Flag-DOP KI mice - mass spec) Su et al. Sci Rep (2017) (HA-DOP - immuno detection) Arttamangkul et al. Mol Pharmacol (2008) (Flag-MOP - immuno detection)

We thank the reviewer for the hint and included the additional literature (page 16).

6) It is the opinion of this reviewer that authors should tune down their conclusion on the existence of MOP splice variants. I do not think that this report convincingly shows that such variants do not exist, nor that if they do, they are expressed at a level that possibly has no physiological significance. In fact, even a very low level of expression in a discrete population of cells might play a significant physiological role. In the whole brain, the total amount might seem too low to have any role, but if concentrated in a specific area/nucleus, who knows?

It is an inherent limitation in any science to prove the absence of something. We have approached the question of alternative MOP isoforms from three different directions and found no evidence for their existence at the protein level, given the technical limitations of each method. We agree that not detecting alternative peptides does not exclude the presence of very small amounts of splice variants, which was mentioned in the manuscript page 14 "..., suggesting that alternatively spliced species present at low level in whole brain extract". This is why we present our results as a "lack of detectable splice variants". Therefore, we believe that we did not overstate our conclusions.

Minor comments:

1) I couldn't find how many separate experiments have been conducted.

The information has been added to the figure legends.

2) Predicted MW for each splice variants should be provided.

MWs for the predicted splice variants are now provided in Table S1.

We would like to take this opportunity to thank the Reviewers and the Editorial Board again for their valuable comments and criticisms, which have helped us to greatly improve our manuscript. The manuscript contains original work and is not under consideration elsewhere. We appreciate the time spent by the Reviewers, and we look forward to hearing whether the manuscript is now acceptable for publication.

Sincerely yours,

A. Kliewer

S. Schulz

REVIEWERS' COMMENTS:

Reviewer #2 (Remarks to the Author):

The authors have done a great job addressing all my concerns. I am convinced that this piece of work represents a significant advance to the field and further expand our arsenal of genetically-modified animal models designed to study the opioid receptors.

In particular, I think that the inclusion of the pharmacological and behavioral characterization of this mouse model is very important.

I do not have any additional comments/concerns